Characterization of carbonate fraction of the Atlantic bluefin tuna fin spine bone matrix for stable isotope analysis

Luque Patricia L. plastra@azti.es 1
Sanchez-Ilárduya María Belén 2
Sarmiento Alfredo 3
Murua Hilario 1
Arrizabalaga Haritz 1
1 Marine Research Division, AZTI Tecnalia , Pasaia , Guipuzcoa , Spain
2 Advanced Research Facilities (SGIker)/ X-ray Photoelectron Spectroscopy Laboratory (XPS)/Faculty of Science and Technology, University of the Basque Country (UPV/EHU) , Leioa , Bizcaia , Spain
3 Advanced Research Facilities (Sgiker) / Coupled Multispectroscopy Singular Laboratory (Raman-LASPEA)/ Faculty of Medicine and Odontology, University of the Basque Country (UPV/EHU) , Leioa , Bizcaia , Spain
Sponheimer Matt
Electronic publication date: 2019 Jul 18
Publication date: 2019
Volume: 7
Electronic Location ID: e7176
Received 2018 Dec 11; Accepted 2019 May 23
Copyright: ©2019 Luque et al.
Copyright year: 2019
Copyright holder: Luque et al.
License: This is an open access article distributed under the terms of the Creative Commons Attribution License, which permits unrestricted use, distribution, reproduction and adaptation in any medium and for any purpose provided that it is properly attributed. For attribution, the original author(s), title, publication source (PeerJ) and either DOI or URL of the article must be cited.
License URL: https://creativecommons.org/licenses/by/4.0/

Keywords: Fin spine, Inorganic matrix, Raman spectroscopy, Thunnus thynnus, XPS, FTIR, Isotope analysis

Funding: European Union’s Horizon 2020 No 753304 This work is part of the SIFINS project that has received funding from the European Union’s Horizon 2020 research and innovation programme under the Marie Sklodowska- Curie grant agreement No 753304. The funders had no role in study design, data collection and analysis, decision to publish, or preparation of the manuscript.

==============================
The mineral component of fish otoliths (ear bones), which is aragonitic calcium carbonate (CaCO3), makes this structure the preferred sample choice for measuring biological carbon and oxygen-stable isotopes in order to address fundamental questions in fish ecology and fisheries science. The main drawback is that the removal of otoliths requires sacrificing the specimen, which is particularly impractical for endangered and commercially valuable species such as Atlantic bluefin tuna (Thunnus thynnus) (ABFT). This study explores the suitability of using the first dorsal fin spine bone of ABFT as a non-lethal alternative to otolith analysis or as a complementary hard structure. The fin spines of freshly caught ABFT were collected to identify carbonate ions within the mineral matrix (i.e., hydroxyapatite) and to determine the nature of the carbonate substitution within the crystal lattice, knowledge which is crucial for correct measurement and ecological interpretation of oxygen and carbon stable isotopes of carbonates. Fin spine sections were analyzed via X-ray Photoelectron Spectroscopy (XPS), Raman Spectroscopy, and Fourier Transform InfraRed (FTIR). The XPS survey analysis showed signals of Ca, O, and P (three compositional elements that comprise hydroxyapatite). The Raman and FTIR techniques showed evidence of carbonate ions within the hydroxyapatite matrix, with the IR spectra being the most powerful for identifying the type B carbonate substitution as shown by the carbonate band in the v2 CO32− domain at ∼872 cm−1. The results of this study confirmed the presence of carbonate ions within the mineral matrix of the fin spine bone of ABFT, showing the feasibility of using this calcified structure for analysis of stable isotopes. Overall, our findings will facilitate new approaches to safeguarding commercially valuable and endangered/protected fish species and will open new research avenues to improve fisheries management and species conservation strategies.

Introduction

Use of otoliths (ear bones) found in teleost fish species is preferred by fishery biologists as they provide time-calibrated archives of isotopic information (δ13C and δ18O ) within the mineral matrix that can be used to address fundamental questions in fish ecology and fisheries science related to migration, habitat use (Campana, 2005; Elsdon et al., 2008; McMahon, Berumen & Thorrold, 2012) and stock structure (Patterson, Mcbride & Julien, 2004). The premise of this approach is that as these biogenic structures grow, they precipitate material (e.g., stable isotopes) that are naturally incorporated into their mineral matrix (aragonite, CaCO3) in direct proportion to the concentration at which they occur in their habitat. As the fish grow, they record the chemistry of their natural habitat as well as other environmental parameters (e.g., temperature, salinity) (Lin et al., 2007; Miller, 2009), thereby linking the hard structure chronology with the chemical record of the fish’s life. This enables a retrospective description to be produced of the environmental history of individual fish over ecological time scales (e.g., Walther & Limburg, 2012; Rooker et al., 2014). However, otolith removal from live specimens requires specimen sacrifice, which may be forbidden in the case of endangered and/or protected fish species. In addition, it is not practical for commercially valuable species such as Atlantic bluefin tuna since it greatly affects the appearance of a fish, diminishing its market value. Alternative hard tissues such as the first dorsal fin spine bone may provide valuable and complementary chemical information, so sampling these tissues represents a non-lethal, minimally invasive sampling method (Zymonas & McMahon, 2006). This makes using fin spines particularly attractive for fish species that cannot be sacrificed for their otoliths such as endangered (threatened) fish or those of management concern (e.g., protected, or commercially valuable whole fish). The term “fin spines” is applied to the anterior-most structural components of fins that are unsegmented, rigid, and more calcified than the soft, segmented “fin rays” (Tzadik et al., 2017). Similar to other bones of the endoskeleton, fin spines are assumed to be composed of three main constituents of bone: organic matter (mainly type I collagen), the mineral fraction being composed of hydroxyapatite (HA), and water (10%wt.) (e.g., Koch, 2007; Rey et al., 2009; Ugarte et al., 2011). As such, the mineral component of bone, Hydroxyapatite (Ca10(PO4)6(OH)2), is a type of calcium phosphate apatite, which is deposited onto collagen fibrils, providing strength to the bone structure and also serving as an ion reservoir (LeGeros, 1981). In biological tissues, this mineral matrix is complex, rarely stoichiometric, and usually calcium-deficient. It accommodates chemical substitution relatively easily, incorporating a wide variety of relatively small amounts of other substituent atoms or groups taken up from the surrounding body fluids during bone metabolism (Mathew & Tagaki, 2001; Matsunaga et al., 2010; Figueiredo, Gamelas & Martins, 2012). As such, the HA structure in bone may contain carbonate CO32− ions that substitute either for phosphate (B-type HA) or hydroxyl (A type-HA) groups within the crystal lattice (LeGeros, 1991; Rey et al., 2009). Although the B-type is the preferential CO32− substitution found in the bone of a variety of species (Landi et al., 2003; Murugan, Ramakrishna & Panduranga, 2006), this substitution remains unexplored specifically in the fin spine bone of teleost species. Although these levels of substitution are small, it has been established that these elements are associated with the properties of biological apatite and play a major role in the biochemistry of bone (Ibrahim, Mostafa & Korowash, 2011). However, our lack of understanding of metabolic pathways, routes of ion uptake, and differential abilities of structures to incorporate elements and stable isotopes should be taken into consideration when using these structures (Campana, 1999). The general assumption that the mineral content of fin spine bone (i.e., hydroxyapatite) is identical to other bones needs further exploration, and a thorough understanding of species-specific skeletal biology may make interpretation of these isotopic records possible. For correct measurement and reliable biological interpretation of the isotopic signal of carbonates contained in the fin spine bone, the required first step is to assess whether or not this carbonate substitution is regular within the HA matrix. The main goal of this study is to characterize the carbonate fraction in the fin spine bone and determine the level and nature of the carbonate substitution within the calcium phosphate apatite, which is the main mineral component of the fin spine bone. This is critical to analyzing and interpreting the carbon and oxygen stable isotope signature retained within the mineral fraction of the fin spine bone. We used Fourier transform Infrared (FTIR) and Raman spectroscopy for the bulk analysis and complemented this investigation using X-ray photoelectron spectroscopy (XPS) as a highly sensitive surface technique.

Materials & Methods

Sample collection

In this study, the first spiniform ray of the first dorsal fin (fin spine hereafter) was collected from fresh Atlantic bluefin tuna caught by commercial bait boats in the Bay of Biscay (North-east Atlantic Ocean). The sample set comprised four individual fin spine samples collected from young specimens of 81, 101, 114, and 128 cm straight fork lengths (SFL).

Fin spine bone preparation

Fin spine preparation and sectioning procedures were performed following the procedures described by Luque et al. (2014). Once the fin spine was removed, the remaining skin tissue was carefully removed with a sharp scalpel, avoiding any damage to the surface of the base of the fin spine. Then, before sectioning, individual fin spines were washed with Milli-Q water and air dried at room temperature. A cross-section of ∼1. five mm thickness was sectioned at a point 1.5 times the condyle base width (1.5Dmax) (Figs. 1A and 1B) with an Isomet low-speed diamond saw (Buehler, Lake Bluff, IL, USA). Additionally, fin spine sections underwent an extra cleaning cycle, by placing them into individual one.five mL vials with ultrapure deionized water (Milli-Q, 18.2 Mohm. cm-3) and using an ultrasonic water bath for 10 min to remove excess organic tissue. Finally, the samples were rinsed again with ultra-pure water and placed in new vials, then allowed to dry in a class 100 laminar-flow hood for 48 h.

Figure 1 Description of the sectioning axis location.

(A) An anterior view of the condyle base and the location of Dmax, which is measured along the imaginary line below the hollows; (B) the lateral view of the condyle base with the location of the 1.5 cutting axis at 1.5Dmax from the same imaginary line.

X-ray Photoelectron Spectroscopy (XPS)

XPS experiments were recorded with a SPECS (Berlin, Germany) system equipped with a Phoibos 150 1D-DLD analyzer and using monochromatic AlK α radiation (1486.7 eV, 300 W, 13 kV). Spectra were acquired in the constant pass energy mode at a binding energy (BE) of 80 eV for survey spectra (step energy 1 eV, dwell time 0.1 s) and 30 eV for high-resolution spectra (step energy 0.1 eV, dwell time 0.1 s) with an electron take-off angle of 90°. The spectrometer had previously been calibrated by using the Ag(3d5/2) line at 368.26 eV. The binding energy scales for the samples were corrected by setting the C(1s) peak of the adventitious carbon to 284.6 eV (other peaks were corrected accordingly) (Wagner et al., 1979). The spectra were fitted with the Casa XPS 2.3.16 software, which models the Gauss–Lorentzian contributions, after background subtraction (Shirley). Concentration values were calculated using the atomic sensitivity factors from Scofield cross-sections and considering the instrumental transmission function and corrections due to differences in the inelastic mean free path.

Raman Spectroscopy

The Raman measurements were conducted with a Renishaw In Via Raman spectrometer, linked to a Leica DMLM microscope. The spectra were acquired with a Leica 50x N Plan (0.75 aperture) lens that yields a spatial resolution of two microns. The instrument used a diode laser exciting at a wavelength of 785 nm (diode laser; Torsana, Skodsborg, Denmark) with a power at the source of 350 mW, the maximum power at the sample being 150 mW. The Raman scattered light was dispersed with a grating of 1,200 lines/mm and was detected by a Peltier cooled charge coupled device (CCD) detector of 576 × 384 pixels, with a spectral resolution of 1 cm−1. In all the measurements, the power of the laser was reduced using neutral density filters to avoid photo-decomposition of the samples (burning). For each spectrum, the CCD detector was open for 20 s, and five scans were made at 10% of the maximum power of the 785 nm laser in the spectral window from 150 cm−1 to 3,200 cm−1. The operation of the equipment was fully software-controlled using WiRE 3.4 software.

Fourier Transform Infrared Spectroscopy (FTIR)

All infrared spectra were measured in transmittance mode on a Jasco 4200 spectrometer by grinding samples with potassium bromide (KBr) powder and then pressing them into a disk. Spectra were acquired over the range 4,000 to 400 cm−1. Each spectrum represented the average of 40 scans at a resolution of four cm−1, in order to provide a good signal to noise ratio. The instrument was controlled by Jasco software that also allowed for the processing of the results.

Analysis of spectral data from Raman and FTIR Spectroscopy

The intensities of the amide I, phosphate and carbonate bands were calculated from IR and Raman spectra by being the baseline for each peak linearly corrected using the same wavenumber limits (see Table 1). These spectral features were used to estimate the mineral/matrix ratio, that is the proportion of the mineral (phosphate band) compared to the organic content (in this case Amide I band) and the level of carbonate substitution, i.e., the carbonate/phosphate ratio. Additionally, crystallinity was determined from the inverse of the full width at half-maximum (FWHM) of the v1 phosphate peak in the Raman spectra (Turunen et al., 2011).

Table 1 Wavenumber limits used in the curve fitting of the bands for Raman and FTIR spectra.

Intensities of Amide I, phosphate and carbonate as the spectral features used to estimate mineral/matrix ratio, carbonate/phosphate ratio and crystallinity.

Wavenumber limits (cm−1)	
Bands	Raman	FTIR	
Amide I	1,590–1,730	1,590–1,730	
Phosphate	930–990	900–1200	
Carbonate	1,055–1,090	850–890	

Results & Discussion

XPS

Elemental composition and functional group identification of fin spine samples of bluefin tuna were accomplished by XPS analysis, thereby obtaining survey scans and high-resolution XPS spectra of the detected elements. Figure 2 shows a typical XPS wide spectrum from the bluefin tuna fin spine sample showing the signals of carbon (C(1s) (284.6eV)), nitrogen (N(1s) (∼399eV)), calcium (Ca(2p) (∼347eV)), oxygen (O(1s) (∼531eV)), and phosphorous (P(2p) (∼133eV)) peaks. The last three comprise the compositional elements of the typical hydroxyapatite (Ca10(PO4)6(OH)2). Nitrogen was not detected in sample BFT2 while for BFT1 it was found close to background noise, with its atomic concentration being around 1.4%. Phosphorous was not detected in either of the two bluefin tuna fin spine samples. Additionally, the high-resolution XPS spectra of C, O, Ca, N, and P were analysed in all tuna fin spine samples and their relative atomic concentrations are given in Table 2. Using sample BFT3 as an example, the high-resolution Ca(2p) core spectrum splits into two peaks identified as Ca(2p3∕2) and Ca(2p1∕2) due to spin–orbit coupling. The main peak Ca(2p3∕2) can be observed at 346.6 eV, while Ca(2p1∕2) appears at 350.2 eV. In the same way, the P(2p) core spectrum splits into two peaks, the main peak P(2p3∕2) at 132.4 eV and P(2p1∕2) at 133.4 eV, in good agreement with phosphate type compounds (Lu et al., 2000; Rey et al., 2009). Meanwhile, the high-resolution O(1s) spectrum was fitted with only one component detected at 530.8 eV. Similar spectra for Ca(2p) and O(1s) were obtained from samples BFT1, BFT2 and BFT4 and for P(2p) from sample BFT4. Two contributions could be distinguished for the high-resolution spectra of C(1s) at 284.6eV (main peak) and 287.5 eV. The first one may have been associated with the C-C/C-H bonding. The other peak at 287.5 eV can be attributed to a bonding between carbon and oxygen. The C(1s) spectra observed for the other samples were quite similar, with the main contribution at 284.6 eV. Nitrogen coming from the collagen protein source of bone was also detected in three fin spine samples at binding energies around 399 eV. Regarding composition, all fin tuna spine samples showed high levels of carbon. Overall, the presence of carbon and nitrogen and the high proportion of oxygen (O/Ca and O/P ratios were higher than expected in hydroxyapatite) in all tuna fin spines analyzed in comparison with the synthetic HA (Lu et al., 2000) is certainly evidence of the presence of collagen Type I, an organic matrix that makes up the protein component of bone and is assimilated from the carbon and nitrogen contained in the protein constituents of a consumer’s diet (Koch, 2007; Rey et al., 2009). Because of the high amount of carbon detected due to the organic matrix of the fin spine bone (>80% in % At rel. in samples BFT1 and BFT2, and ∼60% in BFT3 and BFT4), and considering that XPS is a surface-sensitive technique, the relative atomic percentages of the HA elements such as Ca and P were quite low, 5.9% and 4%, respectively, in the case of BFT3. Moreover, Ca and P concentrations were still lower or not detected in the other three samples, as can be seen in Table 2.

Figure 2 XPS survey spectrum of fin spine (BFT3) showing signals for Ca, P, O (three compositional elements of calcium phosphate “hydroxyapatite”).

Table 2 Spectral features of the detected elements in the different samples.

Binding energy (BE), full width at half maximum (FWHM), percentage of relative atomic concentration (% At. rel.), total relative atomic percentage of the element (% At. rel. Total).

Fin spine sample	Element	Id.	BE (eV)	FWHM	% At. rel.	% At. rel. (Total)	
BFT1	C (1s)	C-C, C-H	284.6	2.1	72.3	82.2	
C 1s	286.9	2.1	5.7	
C 1s	288.5	2.1	4.3	
O (1s)	O 1s	530.6	3.0	15.5	15.5	
Ca (2p)	Ca 2p (3/2)	346.6	2.0	0.6	0.9	
Ca 2p (1/2)	350.0	2.0	0.3	
N (1s)	N 1s	399.0	1.9	1.4	1.4	
BFT2	C (1s)	C-C, C-H	284.6	2.2	79.4	85.2	
C 1s	288.2	2.1	5.9	
O (1s)	O 1s	531.8	2.9	12.5	12.5	
Ca (2p)	Ca 2p (3/2)	347.3	2.2	1.6	2.3	
Ca 2p (1/2)	350.9	2.2	0.8	
BFT3	C (1s)	C-C, C-H	284.6	2.7	48.7	57.9	
C 1s	287.5	2.7	9.2	
O (1s)	O 1s	530.8	2.8	24.6	24.6	
Ca (2p)	Ca 2p (3/2)	346.6	2.3	4.0	5.9	
Ca 2p (1/2)	350.2	2.4	2.0	
N (1s)	N 1s	398.8	2.3	7.6	7.6	
P (2p)	P 2p (3/2)	132.4	2.2	2.6	4.0	
P 2p (1/2)	133.4	2.2	1.3	
BFT4	C (1s)	C-C, C-H	284.6	2.6	54.0	65.0	
C 1s	287.4	2.6	10.9	
O (1s)	O 1s	530.6	3.0	23.1	23.1	
Ca (2p)	Ca 2p (3/2)	346.5	2.0	1.9	2.9	
Ca 2p (1/2)	350.1	2.1	1.0	
N (1s)	N 1s	399.0	2.3	7.1	7.1	
P (2p)	P 2p (3/2)	133.0	2.1	1.3	1.9	
P 2p (1/2)	134.0	2.1	0.6	

Based on the concentrations found for Ca and P, a stoichiometric correspondence to synthetic HA Ca5(PO4)3(OH) can be concluded. For samples BFT3 and 4, the observed Ca/P ratios were 1.50 and 1.53, respectively, close to the nominal value of 1.67 in hydroxyapatite. These slightly low values for the Ca/P ratio have previously been reported (Lu et al., 2000). Additionally, carbonate is commonly found in biological minerals (LeGeros, 1981) and is a basic component of bone that might interfere with the biological interpretation of the data, particularly important for isotopic signals. In the present study, the presence of carbonate-type carbon could not be demonstrated by XPS in any of the fin spine samples. This was probably due to the high amount of carbon from the organic fraction that prevented detection of the low contribution of the carbonate in the C(1s) spectrum, referenced at around 289 eV (Liu et al., 2018). Future work should be devoted to pre-treating the fin spine bone samples with chemicals (e.g., hydrazine) commonly used for removing organic material and/or contamination, in order to eliminate competing sources of C and O stable isotopes associated with organic material and secondary carbonates (Snoeck & Pellegrini, 2015; Pellegrini & Snoeck, 2016).

Raman and FTIR spectroscopy

Mineralogical and compositional analysis of the fin spine bone samples was also performed by means of Raman and FTIR spectroscopy. Both techniques revealed that the spectra were very similar to the spectrum of human bone (Fig. 3) confirming that the mineral fraction of tuna fin spine bone mainly consisted of hydroxyapatite. Overall, the most intense bands arise from the mineral fraction, in accordance with its larger proportion in the composite. Most of the absorptions from phosphate vibrations were clearly observed in the spectra of fin spine. Mineral bands were more conspicuous in Raman spectra than in IR spectra, whereas organic signatures appeared stronger in IR spectra. Relative intensities were slightly different, but most matrices were distinguished in both types of spectra. Thus, detailed spectral assignments for a fin spine bone included vibrational modes for phosphate, carbonate and the organic matrix bands of the amides I, II, III and C-H stretch (Table 3). The main differences were found in the phenylalanine band that is weak in the Raman spectrum (1,004–1,005 cm−1) or absent in the IR spectrum, whereas the amide II in the IR spectrum (1,540–1,580 cm−1) was absent in the Raman spectrum (Table 3). In addition, Table 4 displays the main compositional parameters for each fin spine sample obtained from peak intensities in the IR and Raman spectra. The mineral/matrix ratio obtained in the IR spectra seemed to show higher values in larger specimens (i.e., in body size) whereas no clear pattern was observed in the Raman spectra. Considering that the number of analysed specimens was too low to carry out comparative and correlation analysis, the observed trends should be taken with caution. Turunen and colleagues (2011) stated that this ratio describes the degree of mineralization in the bone and average increases during maturation. As this is beyond the scope of this exploratory study, a deeper quantitative study using a larger sample size including immature and mature individuals is certainly necessary to confirm this.

Figure 3 Raman spectra of bluefin fin spine bone and human bone measured in the same instrumental conditions.

Table 3 Band assignments for the Raman and FTIR spectra of bluefin tuna fin spine bone (BFT3).

The baseline for each peak was linearly corrected using the same wave numbers.

Assigment	Raman band (cm −1)	IR band (cm −1)	
ν (O–H), ν (N-H)	 	3,434	
ν (C–H)	2,884, 2,936	2,958, 2,923, 2,847	
Amide I	1,672	1,660	
Amide II	–	1,565	
δ(CH2), CH2 deformation	1,449	1,454	
ν3CO32−	–	1,416	
δ(CH2)	–	1,337	
Amide III	1,241	1,243	
ν1CO32−ν3PO43−	1,075	1,110, 1,028	
ν3PO43− 	1,030	–	
ν(C–C) Phenylalanine	1,005	–	
ν(C–C) Collagen proline	852	–	
ν2CO32	–	872	
ν4 antisymmetric bending (OPO) 	–	602, 560	
ν4PO43−	581	560	
ν2PO43−	430–450	–	

Table 4 Values of compositional parameters for each fin spine obtained from peak intensities in Raman and FTIR spectra.

			Parameters			
Technique	Sample ID	Body length	Mineral/Matrix (intensities)	Carbonate/Phosphate (intensities)	Crystallinity	
Raman	BFT1	128 cm	3.973	0.102	0.058	
 	BFT2	114cm	7.025	0.093	0.061	
 	BFT4	101cm	4.700	0.108	0.056	
mean ±SD	 	 	5.93 ± 1.594	0.101 ± 0.008	0.058 ± 0.003	
FTIR	BFT1	128 cm	2.945	0.077	 	
 	BFT4	101cm	2.171	0.079	 	
 	BFT3	81cm	1.845	0.126	 	
mean ±SD			2.32 ± 0.565	0.094 ± 0.028	 	

FTIR spectra were also very useful in identifying carbonate (CO32−) substitution in the mineral fraction of the bluefin tuna fin spine bone. As is widely known, the CO32− ion can be allocated within the HA lattice at the hydroxide (type A) or at the phosphate (type B) positions, and these different structural positions result in different IR features. Thus, the type A carbonate, commonly has a doublet band (i.e., at 1,545 and 1,450 cm−1) that corresponds to the asymmetric stretching vibration (ν3) and a singlet band at 880 cm−1 that represents an out-of-plane bending vibration (ν2). Type B shows the strongest IR bands of the CO32− ion at about 1,455, 1,410 and 875 cm−1 (Turunen et al., 2011; Fleet, 2015). This study showed that in the bluefin tuna fin spine bone, the carbonate ions originated in a band at 872 cm−1which was assigned to the v 2 bending vibration (Fig. 4). These findings are certainly in line with what has been identified in biological apatites as the band characteristic of type B carbonates (Figueiredo, Gamelas & Martins, 2012). As a result of that, it was verified that the bluefin tuna fin spine bone resembled type B carbonates. Moreover, the v 3 mode (∼1,454, ∼1,416) was overlapped by strong organic absorption bands. This finding should be taken with caution as it may lead to misinterpretations of bone samples (Termine & Lundy, 1973). In the present study, the type B carbonate substitution was not as clearly observed in the Raman spectra as in the IR spectra. The sharpest band at 1,070 cm−1 (ν1 CO32−) was very close to a component of a phosphate band at 1,076 cm −1 (ν3 PO43−) which is commonly overlapped with the band at 1,110 cm−1 due to the anionic substitution of HPO42− (acidic phosphates), very likely leading to misinterpretation of the spectra. The carbonate/phosphate ratio seems to represent different phenomena in each analytical technique. In the present study, the carbonate/phosphate ratios estimated from the Raman spectra were similar for the three individual bluefin tuna fin spine samples (mean 0.101 ± 0.008 SD). However, in the IR spectra, this ratio was slightly higher in the smallest and youngest specimen, BFT3 (81 cm SFL, aged two years old) (Table 4). Nevertheless, this result should be treated cautiously as we only analyzed fin spine samples from juvenile tuna that were three years old and assumed to be immature. This parameter seems to be a good indicator of the bone turnover and remodeling activity of bone (Isaksson et al., 2010), which remarkably occurs at older ages (Luque et al., 2014) and in mature bluefin tunas, and is linked to the physiology of this fish in relation to its feeding, reproduction, migrating and growing characteristics (Santamaria et al., 2015). In addition, the crystallinity index was similar for BFT1 (128 cm SFL), BFT2 (114 cm SFL), and BFT4 (101 cm SFL) with values of 0.058, 0.061, and 0.056, respectively (mean 0.058 ± 0.03 SD) which indicate a similar maturation state, as expected, considering that the three fin spines were collected from immature juvenile bluefin tuna that were three years old. Nevertheless, it should be considered that such parameters (i.e., mineral maturity and crystallinity) depend on several factors, not only age, but also nutrition and health condition (Figueiredo, Gamelas & Martins, 2012). Since it was beyond the scope of the present study, we suggest further spectroscopy analysis using larger sample sizes and including specimens of different maturity stages to confirm whether spectroscopic data may provide a measure of the maturity of the crystal.

Figure 4 Spectral regions in the FTIR spectrum of bluefin tuna fin spine bone related to Type B carbonate substitution.

Conclusions

The major contribution of this study is the confirmation that the mineral matrix (i.e., the apatite calcium phosphate) of fin spine bone regularly showed carbonate ions that substitute for phosphate (B-type) as carbonate ions were detected within all the ABFT fin spines analyzed. As such, these findings, which are generalizable to other fish species, will allow analysis of the carbon and oxygen stable isotope within the mineral matrix, providing important ecological information on the environmental histories of fish, including stock mixing, movement and dispersal patterns etc. in areas where geographic differences in water chemistry exist. The use of fin spines is advantageous as a non-lethal, minimally invasive sampling method. This makes using fin spines particularly promising for fish species that cannot be sacrificed for their otoliths, such as endangered species (i.e., rare or threatened species) or those of management concern (such as stock, protected, or commercially valuable whole fish), which include ABFT, opening new research avenues to improve fisheries management and species conservation strategies. Regarding analytical techniques, Raman and FTIR seem to be more powerful than XPS in providing relevant information on the carbonate environment and its preferential substitution within the HA lattice. The results indicate that caution is required when performing surface analysis as other skeletal components, such as organic matter, probably affect the surface detection of inorganic carbon from carbonate impurities, leading to misinterpretation of structure/function relationships. When studying the fin spine bone mineral fraction, it is recommended to remove the organic component to avoid any misdetection of components of interest.

Supplemental Information

File S1 Elemental composition of bluefin tuna spine bone acquired by XPS

Click here for additional data file.

File S2 FTIR spectra of BFT1,3, and 5 (wavenumber vs % Absorbance) & Raman spectra of BFT1,2 and 5 (Raman Shift vs Intensity)

Click here for additional data file.

File S3 Raman spectra of bluefin tuna fin spine (Raman Shift vs Intensity)

Click here for additional data file.

The fin spine samples used in this work were collected by AZTI under the provision of the ICCAT Atlantic Wide Research Program for Bluefin Tuna (GBYP). The contents of the paper do not necessarily reflect the point of view of ICCAT or of the other funders. The manuscript benefited from helpful discussions with and technical support from Dr. Gorka Bidegain, Juan Carlos Raposo, Dr. Luis Bartolome, Dr. Olatz Zuloaga, Professor Juan Manuel Madariaga.

Additional Information and Declarations

Competing Interests

Author Contributions

Animal Ethics

Data Availability

Patricia Lastra Luque, Hilario Murua and Haritz Arrizabalaga are employed by AZTI Tecnalia, Marine Research Division.

Patricia L. Luque conceived and designed the experiments, analyzed the data, contributed reagents/materials/analysis tools, prepared figures and/or tables, authored or reviewed drafts of the paper, approved the final draft.

María Belén Sanchez-Ilárduya and Alfredo Sarmiento performed the experiments, analyzed the data, contributed reagents/materials/analysis tools, prepared figures and/or tables, authored or reviewed drafts of the paper.

Hilario Murua and Haritz Arrizabalaga authored or reviewed drafts of the paper, approved the final draft.

The following information was supplied relating to ethical approvals (i.e., approving body and any reference numbers):

AZTI facilities are licensed for collection of samples according to the Spanish legislation (RD1201/2005, royal law for the protection of animals used in scientific experiments).

There are no special ethical issues associated with this work. This study uses fish spine samples collected from fish captured by commercial fisheries of Spain. However, all project activities are undertaken within the clear boundaries of national and EU legal frameworks, specifically those relating to animal welfare (i.e., Directive 2010/63/EU).

The following information was supplied regarding data availability:

The raw data are available as Supplemental Files. The raw data show elemental composition and functional groups identification_ XPS data of BFT 1,2,3,4, FTIR spectra of BFT 1, 3 & 5 (wavenumber vs. %Transmitance), and Raman spectra of BFT 1, 2 & 5 (Raman Shift vs. Intensity), respectively. Raw data were used to build the spectrums provided in Figs. 2–4.

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
