# Peer review of "Characterization of carbonate fraction of the Atlantic bluefin tuna fin spine bone matrix for stable isotope analysis"

_PeerJ, doi:10.7717/peerj.7176_

## Round 0.1 · original submission · Major Revisions

Please revise your manuscript in light of the comments of both reviewers. Pay especially close attention to the comments in the Experimental Design Section and address those in some detail in your response.

·

Basic reporting

The work is about the characterization of bluefin tuna spines using innovative techniques in the field such as electronic X-ray spectroscopy (XPS) and Raman spectroscopy. It consists of simple experiments without sacrificing the life of the specimen. With these techniques, data are obtained to provide information in a way that can be used to a better understanding of the protected species.
Despite its simplicity and low number of samples, it can provide a new path for the analysis of the environment in which these species develop. The structure and the presentation of figures and tables seems appropriate. The English used in the paper should be subject to a slight revision.

Experimental design

No comments.

Validity of the findings

In my opinion, the article presents the application of three spectroscopic techniques for the characterization of biological samples that are not ordinary in this field of science, therefore can contribute with an important advance for the field. With simple experiments to carry out, very promising results are obtained.
From my point of view, the only aspect to improve is the number of samples to be analyze. On the other hand, the authors are aware of the fact and highlight this issue in the text.

Additional comments

I will raise some questions about the work:
On line 207, when speaking about XPS techniques for nitrogen contents detection, the origin of nitrogen should be mentioned.
A comment on why P does not appear in two of the samples is missing.
In Figure 4, the 2p transition of phosphorus should be deconvoluted with two contributions, that is, as 2p1 / 2 and 2p3 / 2.
In Table 1, for a better understanding of the table contents it should be included some lines for dividing contents of the different samples
In Table 2, in the first column (assignment) the line must say v (C-H).

·

Basic reporting

One of the main problems of this paper is the language. A review by an English native speaker would really make a difference and would allow for the paper to become more accurate and comprehendible. I have made a few suggestions within the manuscript (given below), but I strongly believe that a detailed review should be considered.

Line 43: “for a correct” should be replaced by “for the correct”

The abstract should be written in the third person and not the first, for example in Line 44: “We collected the fin spines from freshly caught bluefin tuna (Thunnus thynnus) ” should be replaced by “The fin spines of freshly caught bluefin tuna (Thunnus thynnus) were collected..”
Line 46: “Raman Spectroscopy” instead of “Raman”, and “spectroscopy” instead of “spectroscopies”.
Line 47: “positions” instead of “areas”?
Line 50: “the C(1s)” instead of “C(1s)”
Line 51: Please rephrase the whole sentence…for example: In Line 52: “allowed for the study of the carbonate species “ instead of “detect the presence of carbonate ions”, in Line 52: “with the IR spectra being the most” instead of “being IR spectra more ”, in Line 53: “the HA lattice” instead of “HA lattice”, etc…
Line 54: “The results of this study” instead of “Findings of this study”
Line 55: the word “certainly” should be deleted and lines 56-57 should be rephrased in order to be more comprehendible.
Line 63: “otoliths, fin spines” instead of “otolith, fin spine”
Line 65: “McMahon” instead of “Mcmahon” and “migration” instead of “movements”?
Line 67-69: Rephrase, it is a very confusing sentence
Line 71: “parameters” instead of “conditions”
Line 75: “isotopes in” instead of “isotopes on”
Line 80: “its” instead of “their”
Line 81: rephrase “a minimally invasive hard structure”
Line 83: Zymonas and McMahon, 2011 does not appear in Reference list, while “Zymonas, ND,McMahon TE. 2006” from reference list does not appear in text.
Line 85: what does “most structural components” mean?
Line 91: I do not understand the sentence” The hydroxyapatite (HA) ((Ca10(PO4)6(OH)2), is commonly used as a model for describing the inorganic components of bones and teeth due to its close chemical resemblance to apatite (LeGeros, 1981)”. Hydroxylapatite IS a type of apatite. Please rephrase. Once again, this has to do with the use of the language.
Line 93: “In biological tissues” instead of “As they occur in biological tissues”
Line 98: “bone conatins” instead of “bone can contain”
Line 117: “comprised of” instead of “comprised”
Line 121: “procedures” instead of “procedure”
Line 122: perhaps “removed” instead of “extracted”
Line 124-127: rephrase please
Line 128: “by placing” instead of “placing”
Line 129: delete “cleaning”
Line 130: I do not understand the last phrase.
Line 132: “in a class” instead of “in class”
Line 153: “with the maximum power at the sample being 150 mW” instead of “being the maximum power at the sample 150 mW”
Line 154: “detected by a Peltier cooled charge coupled device (CCD) detector with 576 × 384 pixels” instead of “the detection is done by a Peltier cooled charge coupled device (CCD) detector with 576 × 384 pixels”
Line 162: “were measured in” instead of “collected in this work have been carried out in” and “mode on” instead of “mode in”
Line 164: “Spectra were acquired over the range 4000 to 400 cm-1 while each spectrum represents the average of 40 scans at a resolution of 4 cm−1, in order to provide a good signal to noise ratio” instead of “Spectra were acquired from 4000 to 400 cm-1 with a spectral resolution of 4 cm-1 and accumulating 40 scans in each spectrum to get a good signal to noise ratio”
Line 166:” allows for the processing of the results” instead of “allows processing the results”
Line 168: “Analysis of spectral data from Raman and FTIR Spectroscopy” instead of “Analysis of spectral data in Raman and FTIR”
Line 169: “The positions” instead of “The areas” and “of the amide” instead of “of amide”
Line 171: “from the IR” instead of “from IR”
Table 2 (in line 171) is mentioned in the text before Table 1 (line 194), please correct according to the journals instructions.
Line 172: what does “was linearly corrected using the same wavenumbers” mean? Please clarify.
Line 174: “from the ratio of the integrated” instead of “from the integrated”
Line 173-177: More detail must be provided concerning the specific integrated areas used (exact areas used for every ratio in wavenumbers, perhaps also a graph indicating the exact areas used).
Line 179: References
Line 189: “…peaks. The last three comprise the compositional elements of” instead of “…peaks, comprising the three later compositional elements of”
Line 190: “sample BFT2”instead of “BFT2 sample”
Line 191: “, with its atomic concentration being around 1.4%” instead of “, being its atomic concentration around 1.4%.”
Line 191-192: “Phosphorous” instead of “Regarding to phosphorous, it” and “the” instead of “these”.
Line 194: “are given” instead of “resumed”
Line 195: The brackets should be deleted.
Line 201: “from sample” instead of “in sample”
Line 208: “overall” can be deleted, as it used again in the next sentence
Line 212: “probably due” instead of “due very likely”
Line 216: “low” instead of “small”
Line 217: “in Table” instead of “in the Table”
Line 218: “Based on the concentrations” instead of “Analyzing the concentrations”
Line 218-219: “the stoichiometric correspondence to synthetic HA Ca5(PO4)3(OH) can be concluded” instead of “it can be readily seen the stoichiometric correspondence to synthetic HA Ca5(PO4)3(OH).”
Line 219 onwards: usually the Ca/P ratio is calculated and not the opposite, as mentioned here. Even Lu et al, 2000 which is cited here, uses the Ca/P ratio and not the opposite. I suggest the correction of this ratio. It would be then comparable to relevant studies from the bibliography.
Line 220: “close” instead of “closed” and “ of 0.6 for hydroxyapatite“ instead of “0.6 in hydroxyapatite”
Line 222-223: “is a basic component of bone” instead of “is one such constituent regularly observed in bone”
Line 224-228: Please rephrase..
Line 230: “in order to eliminate” instead of “eliminating hence”
Line 236: “spectroscopy” instead of “spectroscopies”
Line 243: delete “bands”
From this point onwards I will not point out the language problems, as they are too many. Please ask an English native speaker to go through the manuscript with you.
Line 279-280: Which peak are the authors referring to? Which peak in IR spectra combines all types of carbonates?
Line 287: Could the authors explain this ? What do they mean by more mature mineral?
Line 288: Since crystallinity is measured by Raman, it could also be measured by FTIR according to Weiner & Yosef, 1990. This way it could comparable to other relevant studies.
Line 289: I don’t see how the crystallinity measurements help this study. Perhaps if they were compared to other similar measurements (other studies) or to fresh bone. Here they only prove the obvious, that the samples are all of similar maturity.
I am not sure if Figures 4-7 are necessary. Does fig.3 combined with Table 1 not provide the necessary data?
Figures 6, 7, 8, 9 are not mentioned within the manuscript. On the contrary, Figurer 3a, 3b, 3c, 3d etc are mentioned within the manuscript but are not given as figures. They obviously correspond to figures 6-9. Also, in Figure 8, a human bone sample is used for comparison. This is not mentioned/described within the material chapter. Please include the necessary information.
Table 3: “Parameters” instead of “paremeters”, “crystallinity” instead of “crystallinity”. Also, how are the mineral/Matrix (areas) and mineral/matrix (intensities) used in this paper? How do they help? I cannot find any discussion/conclusions relevant to these ratios.
The specific peaks used for the calculation of the Mineral/Matrix and Carbonate/Phosphate ratios should be given in detail (here and within the text)
References
1) In text citations should be double checked, as they do not fully comply with the journals indications. For example, for three or fewer authors, all author names should be listed:
For example:
Line 66: “Acosta-Pachón, Ortega-García & Graham, 2015” instead of “Acosta-Pachón et al., 2015”
Line 97: “Figueiredo, Gamelas & Martins, 2012” instead of “ Figueiredo et al., 2012”
Please check the rest of the text.
Also “Walther and Limburg, 2012” should be written “Walther & Limburg, 2012”
2) 377-378: “LeGeros” instead of “Legeros”.
3) Line 388: insert space between references
4) Line 447 onwards: change text size from 11 to 12, to comply with the rest of the manuscript.
5) In the reference section, the format should be checked and corrected according to the journal’s instructions. For example:
Landi E, Celotti G, Logroscino G, Tampieri A. 2003. Carbonated hydroxyapatite as bone substitute. Journal of the European Ceramic Society 23(15): 2931–2937 doi.org/10.1016/S0955-2219(03)00304-2.
Instead of:
Landi, E., Celotti, G., Logroscino, G., and Tampieri, A. (2003). Carbonated hydroxyapatite as bone substitute. J. Eur. Ceram. Soc. 23(15), 2931–2937. https://doi.org/10.1016/S0955-2219(03)00304-2.
Please check the rest of the text.
6) The following references are listed in the References chapter but cannot be found within the text:
Farlay, D., Panczer, G., Rey, C., Delmas, P. D., & Boivin, G. (2010). Mineral maturity and crystallinity index are distinct characteristics of bone mineral. Journal of Bone and Mineral Metabolism. 28(4), 433–445. https://doi.org/10.1007/s00774-009-0146-7.

Gamsjaeger, S., Masic, A., Roschger, P., Kazanci, M., Dunlop, J. W. C., Klaushofer, K., et al. (2010). Cortical bone composition and orientation as a function of animal and tissue age in mice by Raman spectroscopy. Bone. 47(2), 392-399.

Experimental design

This paper describes the study of fin spine bones of fresh blue tuna, by Raman and IR Spectroscopy combined with XPS. The objective of this work is to prove that fin spine bones can be used for isotopic analyses of tissues of such fish, especially when they include endangered species, instead of other parts of the fish (otoliths etc) that can only be taken from killed individuals.
In the abstract but also throughout the manuscript, it appears that the main result of this study is the verification of the fin spine bones composition. I am afraid that this is not something new, as the composition of the bio-mineral of bone (fish bone as well) is well known. I believe it should be rewritten in order to better indicate the scope of this study, perhaps as it is analyzed in the last paragraph of the introduction.
In general, I do not see how this paper contributes to our general knowledge of bone bio-apatite, further than simply verifying what we already know in general, for the specific kind of bone (fin spine bone). Obviously, the authors want to suggest the use of this kind of bone instead of other parts of the fish usually used for isotopic analysis. I am afraid that this is not stressed enough in the text. I believe that the objectives of this paper should be explained in a different manner, so that its importance for fisheries studies is stressed.

Validity of the findings

The presentation of the results could be improved, in order to better serve the original scope of this paper, as suggested in the first part of this review.

---

## Round 0.2 · Minor Revisions

You have done a good job addressing the scientific concerns. However, the language needs work before this could be published.

Please have a colleague who is fluent in English, or a professional editing company, go over the manuscript to correct the language.

---

## Round 0.3 · accepted · Accept

Thank you for the changes. It reads better now.

#